# Evidence-based conservation education in Mexican communities: Connecting arts and science

Montserrat Franquesa-Soler[1,2¤]*, Lucía Jorge-Sales[2], John F. Aristizabal[1,2], Patricia Moreno-Casasola[3], Juan Carlos Serio-Silva[1]

1 Red de Biología y Conservación de Vertebrados, Instituto de Ecología AC, Xalapa, Veracruz, México,
2 Primate Conservation and Sustainable Development, Miku Conservación AC, Xalapa, Veracruz, México,
3 Red de Ecología Funcional, Instituto de Ecología AC, Xalapa, Veracruz, México

¤ Current address: Laboratorio de Interacciones Planta-Animal, Instituto de Investigaciones en Ecosistemas y Sustentabilidad, Universidad Autónoma de México (IIES-UNAM), Morelia, Michoacán, México
* franquesamontse@gmail.com

**Data Availability Statement:** All relevant data are in the paper and its Supporting Information files.

**Funding:** We thank Consejo Nacional de Ciencia y Tecnología (CONACyT, Mexico) for offering the

## Abstract

Several studies suggest that 63% of primate species are currently threatened due to deforestation, pet-trade, and bushmeat hunting. Successful primate conservation strategies require effective educational programs capable of enhancing critical system-thinking and responsible behavior towards these species. Arts-based conservation education can simultaneously foster cognitive and emotional processes. In this paper, we evaluate an arts-based educational program focused on the conservation of black howler monkeys (*Alouatta pigra*). Our goals were to determine (1) whether children's knowledge changed with our educational techniques, (2) if there was a particular educational technique that better improved the children's learning, and (3) the children's emotional feedback regarding the whole program. A total of 229 children from communities located in primate-habitat areas, both inside and outside protected areas, participated in the study. Different educational techniques were tested (storytelling, theater and shadow puppets), contrasted with a control group, and evaluated through an analysis of drawings. Our results showed that children's knowledge increase with each art-based technique, with storytelling being the most effective for children's learning. Specific drawings indicators also revealed the increase of children's knowledge and a decrease of misconceptions between pre and post evaluations. Finally, a satisfaction survey about the program showed a high positive feedback. The study highlights the value of designing multidisciplinary projects, where arts-based education program (grounded in scientific information) has shown to be a successful way to communicate animal knowledge and promote conservation.

## Introduction

Science and art are not opposed activities, they are both human concepts related to the creative processes. The ability to imagine the unimaginable is a prized attribute for both artists and scientists. Education on arts is an education focused on professional and vocational guidance for

scholarship to MFS (N° 556384), Instituto de Ecología, A.C (INECOL, A.C, Mexico), DGAPA-UNAM Postdoctoral Fellowship, and to Juan Carlos Serio-Silva for financial support.

**Competing interests:** The authors have declared that no competing interests exist.

a specific art [1], while education through arts considers art as a learning vehicle for other subjects and as a means to achieve more general educational outcomes [2]. Current research advocates that integrating art and science education could engage learners into creating ingenious projects and inspire them to articulate science in different ways [3]. Nevertheless, the integration of arts in everyday science teaching has been little explored, especially in the field of biodiversity conservation [4]. Here, we present a case study in which an approach of education through arts was used to achieve primate conservation education within a formal schooling system in southeastern Mexico.

Primates are not only crucial for ecosystem functioning, due to seed dispersal services [5,6] and herbivory, as the often make up a large proportion of the vertebrate biomass in tropical forests [6]. They also play important roles in many aspects of human societies (culture, religion, livelihood), and are key for our understanding of human evolution, biology and behavior [7]. Nevertheless, many primates are threatened with extinction (63%) [8], with the primary threats being habitat loss, pet-trade, and bushmeat hunting [7, 9]. Mexico is the northernmost distribution for New World primates with two of their native species *Alouatta pigra* and *Ateles geoffroyi* categorized as 'Endangered' by the IUCN [10].

Because most primates live in tropical regions, where low-income nations predominated, primate conservation programs usually take place in socio-economic contexts characterized by high poverty levels, limited funding opportunities, political instability and corruption [11]. In the face of these challenges, successful primate conservation requires a multidisciplinary approach that needs to be nourished by theory and practice from, at least, the fields of biology, anthropology, psychology, economics, and education [12]. Then, integrating the natural and social sciences might influence positively that decision-making during planning, implementation and management are guided by the best available information [13].

Conservation efforts should also encourage local participation and incorporate local knowledge systems to inform culturally-relevant educational programs that inspire respect towards primates and their habitats in culturally-relevant ways [14,15]. Additionally, these animals are charismatic species, which facilitates the implementation of conservation strategies [4, 16, 17], helping to reach wider goals in biodiversity conservation. However, the strategy of charismatic and flagship species is more effective when target species have links to peoples' cultural identity [16]. Furthermore, as a component of multidisciplinary conservation efforts, appropriate education and outreach programs could promote sustainable behavior (e.g. declining poaching levels), and guide decision and policy-making with and natural resources [18].

Conservation Education (CE) is a crucial component in the process of solving current environmental problems through its role in increasing awareness and modifying attitudes of the general population. Also, CE promotes that researchers and practitioners acquire important knowledge and skills necessary for advancing conservation goals [19]. Nevertheless, there is a need for seeking an interdisciplinary CE in order to be able to tackle complex, multi-disciplinary environmental challenges. To design an efficient CE program, i.e. one that promotes meaningful and transformative learning, it is necessary to consider context-specific factors such as age and learning strategies [20]. In Mexico, CE has been mainly dominated by a traditional education perspective, due to the influence of the fields of biology and technology. In these fields, information is usually transmitted through teaching methods designed for receptive and passive learning, reinforcing a single area of human development: the cognitive domain [23, 24]. However, the emotional domain is needed for integrated child development (e.g. creativity, critical thinking) and for a harmonious relationship with nature [20]. In addition, many children seem to have lost enthusiasm in nature because they perceive it as less attractive than social media or electronic games. Hence, we need to seek for innovative methods to (re)awaken and (re)nourish the sensibility of children and re(build) a new relationship with the outdoors [21].

The emotional perspective on conservation behavior needs to be included on the level of the design and performance of the educational intervention programs to result in long-term changes in feelings and behaviors [22]. For this reason, artistic and creative approaches can facilitate affective knowledge, as well as deepen the emotional connections between people and places [23], thereby maximizing all senses (auditory, visual, kinesthetic, etc.).

It is also vital to include CE in formal education, where the school becomes the cornerstone in promoting values about the society-environment relationship and fostering a critical thinking to face the different scenarios of environmental problems. Besides, in the Mexican rural context, the school is a central point of connection with the rest of the community. The future decision-making group is in the classroom today and the current decision-makers have relatives at the school too.

On the other hand, the benefits of arts education in elementary education are numerous, such as strengthening self-esteem, stimulating creativity and learning, and other moral values that are not acquired with compulsory subjects (taught them in a traditional way). However, in Mexico there is a weak presence of art in schools. Of the 800 hours of classes that are taught annually in primary school, only 40 hours are dedicated to art [24,25]. An accurate assessment of CE programs outcomes would allow to identify specific roles that art can play in biodiversity conservation. Conservation issues can only be solved with creative and critical thinkers, different ways of perceiving and caring about the world should help us to conserve it [20].

### Primate conservation education

CE programs have been reported to change people's perceptions, knowledge and behaviors; thus, they are considered a key element of primate conservation initiatives [12]. Nonetheless, in indigenous communities it is important to consider different approaches from the post-development concepts (e.g. *buen vivir*, *ubuntu*) in order to take into account the traditional knowledge and different worldviews and belief-systems [26].

In practice, primate CE programs vary, facing several difficulties depending on the social and cultural context, encompassing different audiences, having dissimilar lengths of time, and employing a variety of methods including active and passive learning strategies such as nature clubs, documentaries or comic books [27, 28, 29, 30]. Despite there being recent projects incorporating some artistic activities in primate CE, systematic evaluations of effectiveness of these activities remain few or focus mainly on the verbal domain (e.g. questionnaires) as opposed to drawings analysis or photo elicitation [13]. Moreover, in Mexico, there is an overall lack of evaluation of primate CE programs, none of which use artistic approaches. This study seeks to fill this gap by employing an arts-based approach to primate conservation education and evaluating its effectiveness. In this paper, we designed and applied an arts-and-science based educational program in formal schools from southern Mexico in order to evaluate the effectiveness of different teaching strategies in conveying the importance of the black howler monkey (*A. pigra*). Particularly, we aimed to determine (1) if there is a change in children's knowledge caused by an educational intervention, (2) which is the most effective intervention technique to improve the children's knowledge (i.e. learning), and (3) the children's perceptions and feelings concerning the whole educational program, in order to provide spaces for giving them a voice.

## Material and methods

### Study site and subjects

We conducted the study in 12 different communities throughout southern Mexico (Fig 1) during an annual elementary school cycle (2015–2016).

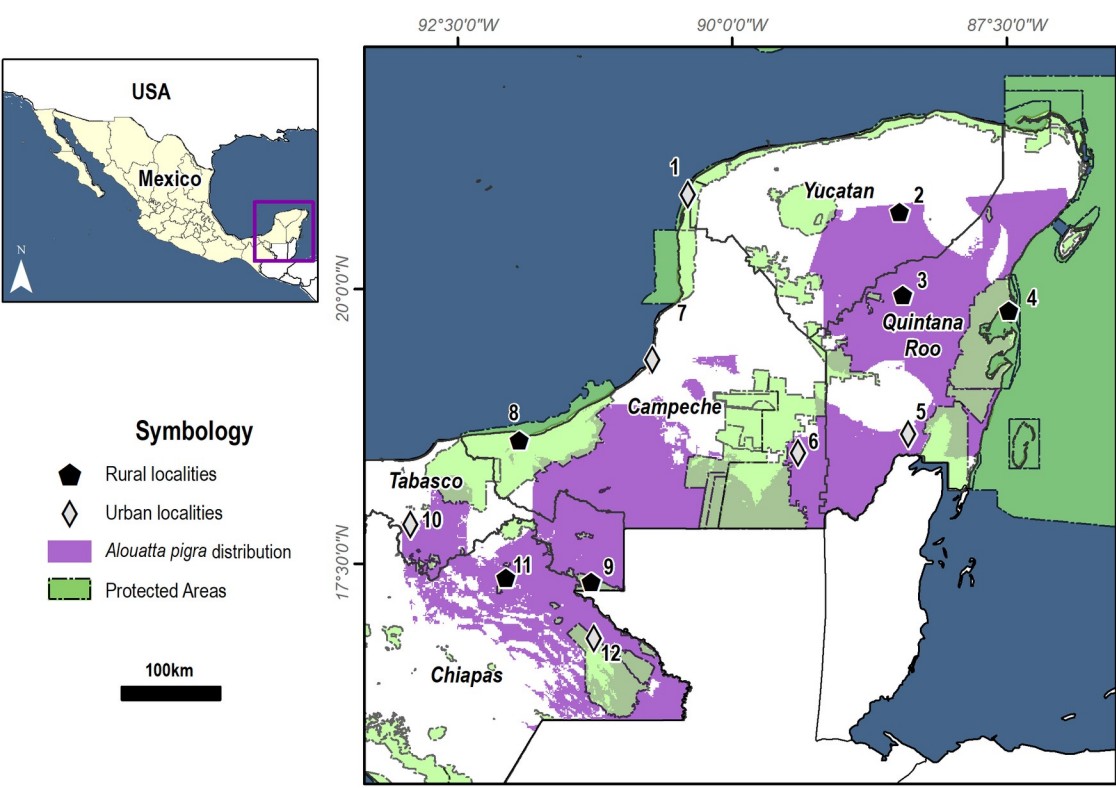

**Fig 1. The area where black howler monkeys are distributed and the communities that were randomly selected.** The 12 communities (6 rural and 6 urban) were within Natural Protected Areas. 1: Celestún, 2: San Francisco Tinum (Yucatán); 3: Dzoyolá, 4: Punta Allen, 5: Bacalar (Quintana Roo); 6: Xpujil, 7: Champotón, 8: Puerto Rico (Campeche); 9 Redención del Campesino, 10: Playas del Rosario (Tabasco); 11: Jerusalen, 12: Nueva Palestina (Chiapas). * Digital cartography used in this Figure is distributed under a Creative Commons Attribution-Noncommercial 2.5 license from Geoportal Conabio.

Study sites were selected based on the geographical distribution of the black howler monkey (*A.pigra*), which corresponds to approximately 250,000 km$^2$ in the states of Tabasco, Chiapas, Campeche, Yucatan, and Quintana Roo. Communities were randomly chosen from the INEGI database [31], with selection criteria being rural or urban (less or more than 2,500 people, respectively) and location inside or outside a protected area (PA). The region is characterized by a strong indigenous heritage; there are around seven million Maya people that still live today in Mexico and Guatemala, many of them are native speakers of Mayan and their variations (e.g. Ch'ol, Tzeltal, Tzotzil, Yukatek) rather than Spanish [32]. A total of 229 students aged 8–10 years from 12 primary schools participated in this study. We considered three factors that could potentially affect our results: gender (boys 48.9% vs. girls 51.1%), context (rural 48.9% vs. urban 51.1%), and PAs (inside PAs 43.7% and outside PAs 56.3%). Our educational intervention was done with the official permission of the *Secretaría de Educación Pública-* Secretariat of Public Education in the Government office of each State, the School Council and the Municipal Comissioner. This study was also approved for the Ethical Committee of the Instituto de Ecología, AC.

### Intervention design

For conducting and evaluating the arts-and-science based educational program at the selected schools we followed a sequential intervention that consisted of three stages at each school:

**Table 1. Description of all the arts-based educational techniques used, including previous activities and materials used.**

| Technique and sense | Previous activity | Description | Material |
|---|---|---|---|
| **Storytelling: Auditory** | Listening to environmental sounds and recreating them. | Storytelling performed by the artistic facilitator, with some sounds through speakers. Students were blindfolded to enhance listening. | Speakers and scarfs. |
| **Shadow puppets: Visual** | Guiding dog game and observational games of the natural environment. | Vision was the predominant sense in this educational technique, where the story was told through shadow puppets performed by the artistic facilitator, supported by text posters and sometimes accompanied with environmental sounds. | Homemade wooden puppet theater with wheels, styrene puppets and speakers. |
| **Theater: Kinesthetic** | Physical and theater warm-up games. | First reading the story aloud and ensuring comprehension. Then, performing the story using only their imagination, body language and props. | Story printed for team work and props (balls and scarfs). |
| **Control group: Mix** | Physical warm up games and a dance activity while music was playing. | Future scenario activity about their community. Children were asked to draw a specific place of their community in the present and 50 years in the future. | Sheet of paper and colored pencils. |

1) Rapport-building phase, via ice breaker games for creating a trusting environment, and pre-evaluation of knowledge through the analysis of drawing contents [14], 2) intervention, four groups per school participated, three of which were part of the arts-based educational techniques, with the fourth group serving as the control group; 3) post-evaluation of knowledge (through drawings analysis), and reinforcement of learning, carried out one month after the intervention. Before the intervention and in parallel, meetings were held with the parents and teachers to explain the activities that we had planned, providing opportunities for them to share feelings and suggestions, and give their verbal consent as legal tutors of the research participants. Parents also signed two permits, one allowing their children go outside the school if the activity required it, and the second giving us permission to use their children's images in scientific or non-scientific publications, as well as press and social media.

## Arts-based educational techniques

The intervention was carried out following different artistic expressions, each focused on a specific sense: storytelling (auditory sense), shadow puppets (visual sense), and theater (kinesthetic sense) (Table 1). Each educational technique had the same duration (20–30 min), and the same content about black howler monkeys (geographical distribution, basic behavior, ecology, conservation, and traditional knowledge), but varied in the form and style in which it was communicated. Activities held for the control groups were not related to the black howler monkey. Instead, a future scenario drawing activity was carried out in these groups, where children were asked to draw their community as they see in the present and what they imagine it would be like in 50 years. Previous to the main activities, we conducted ice breaker games and artistic exercises in order to prepare the artistic language for each education technique (auditory, visual and kinesthetic).

## Evaluation of drawings: Creative and colorful data

Drawings were used to measure the children's knowledge about black howler monkeys (pre and post) [14,33,34]. Crayons and a sheet of paper with a howler monkey silhouette were given to the children, and they were encouraged to make a drawing that could answer the following question 'What does this animal need to live well?' (Fig 2). There was no discussion before starting the drawing session, except to introduce the activity. Students were given 50–60 minutes to complete the drawings.

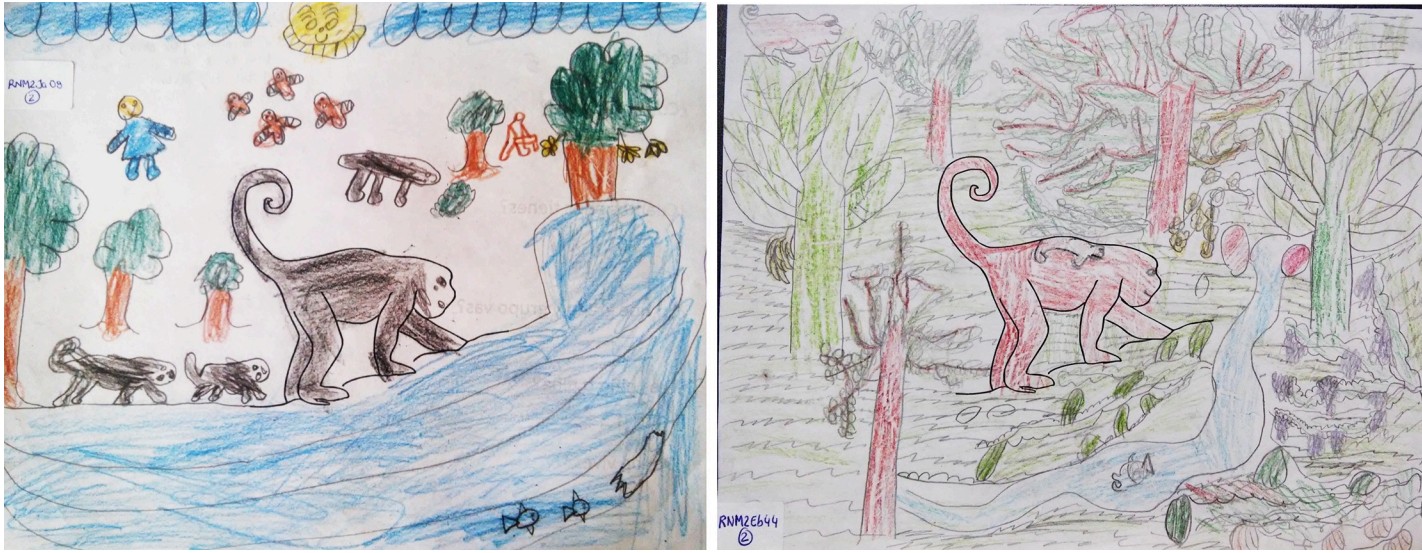

**Fig 2. Some of the children's responses to the question "What does this animal need to live well?", through drawings.** They were used to analyze children's knowledge.

We asked teachers not to interact with children, discuss their drawings or provide additional explanation during the activity. Also, no books or images were allowed during the drawing activity. After drawings were completed, we asked a series of questions to each individual to clarify the objects and actions depicted in them, to improve our understanding of the children's perceptions. To minimize bias during interpretation, all drawings were analyzed by the same pair of evaluators, each one from different disciplinary backgrounds (i.e., conservation biology and psychology) but who were familiar with the monkeys and the children's culture. Both evaluators took part in the classroom activities. Strict pre-determined rules of interpretation were followed (see Franquesa-Soler & Serio-Silva [14] for methodological details). Inter-rater reliability was measured by calculating the percentage of agreement. The two evaluators had 83% of agreement (values > 80% are considered reliable [35]).

## Assessing children's knowledge

Drawings were used to evaluate the effectiveness of each method of the arts-based educational techniques (storytelling, shadow puppets and theater) in expanding students' knowledge about black howler monkey behavior, ecology, and conservation. We assessed students' knowledge about black howlers in terms of the following categories: 1) howler fur color–with black being the correct color, 2) tree–at least one tree appears in the drawing, 3) canopy–locating this animal on a tree or branch), 4) food–any type of correct food source they can imagine this animal needs, 5) family–other howlers in the drawing, and 6) other–e.g., representations of other activities such as playing, sophisticated knowledge about the species, conservation messages. For each of these categories, we assigned a knowledge score of 0 (absence) or 1 (presence). Pre and post scores were compared to assess the effectiveness of each method of intervention (see below). In addition, the following specific indicators of the change between the pre and post were considered: (i) yellow mombin (*Spondias mombin*), also known locally as *jobo*, is a fruit eaten by black howler monkey and it was included in the content of the stories, so we evaluated the appearance of this element in the post-drawings; (ii) assessment of whether several assumptions or misconceptions that were detected in the pre-analysis (banana as food source,

brown color to describe this primate species, and locating the animal on the forest floor [14]) persisted after the interventions.

## Children's perception and feelings

After the educational technique took place, the facilitator talked to the children about their perceptions, and to find out what they felt and understood about the activity, to reinforce the learning process. Finally, we gave the children an opportunity to express one of their favorite's moments through another drawing. This activity was optional.

After intervention was finished, a satisfaction survey was used to identify children's perceptions and suggestions about our performance. Each participant was asked to answer eight questions about the program (i.e., How much did you learn? Could you give your opinion? Do you think this topic is important? Did you like how we told you the story? Would you repeat the experience? Did you have enough time to learn? Did you have fun? Did you like working in a team?). To answer them we used a dartboard prototype with three different code colors: green–very satisfied, yellow–v medium, and red–not satisfied. Also, we hung a cardboard on the wall with the title "How did I find the experience" and encouraged the children to write down their thoughts and feedback about the activities, our presence, or a special moment they wanted to share (Fig 3).

## Data analysis

The change of children's knowledge between pre and post intervention (storytelling, shadow puppets, theater and control) was evaluated with U Mann-Whitney Test. The effectiveness of each educational technique on children's knowledge, measured by the pre-post score difference as a magnitude for children's learning was evaluated with GLMM [36]. In the model, educational technique, context (rural or urban), gender (boy or girl) and location (inside or outside PAs), were considered predictor variables (fixed factors). Given that children in a school were surveyed twice, school was included as a random factor. The fit of the full model was compared with a null model that included only the intercept and random term, using maximum likelihood. We set a ΔAIC threshold of five to consider a model significantly better than the null model [37]. GLMM was run using the package lme4 [38] within the statistical program R (version 3.2.0) [39]. Finally, to assess students' satisfaction with our intervention, we calculated percentages for each level of satisfaction (i.e., green–very satisfied, yellow–medium, and red–not satisfied) from the dartboard exercise.

## Results

### Children's knowledge

We found that children's knowledge increased with educational techniques, but not with the control group (Fig 4). When comparing pre vs. post scores, the storytelling technique showed the largest increase in knowledge (Mann–Whitney U-test: $U = 1009$, $Z = -4.9$, $P < 0.001$), followed by theater ($U = 770$, $Z = -41.3$, $P < 0.001$) and shadow-puppets ($U = 1167$, $Z = -3.4$, $P < 0.001$). Of all factors considered in the GLMM, only technique and context had an effect on the pre-post score difference (Table 2). Storytelling was the best technique, when compared to the control group, for learning about howler monkey behavior, ecology, and conservation. Also, the urban context had a slight effect. (*See parameter in* Table 2).

Full model: Score (learning)~ **Technique** + **Context** + Gender + Location (variables in bold were significant). ΔAIC threshold of minimum model = 6. Storytelling effect is compared with

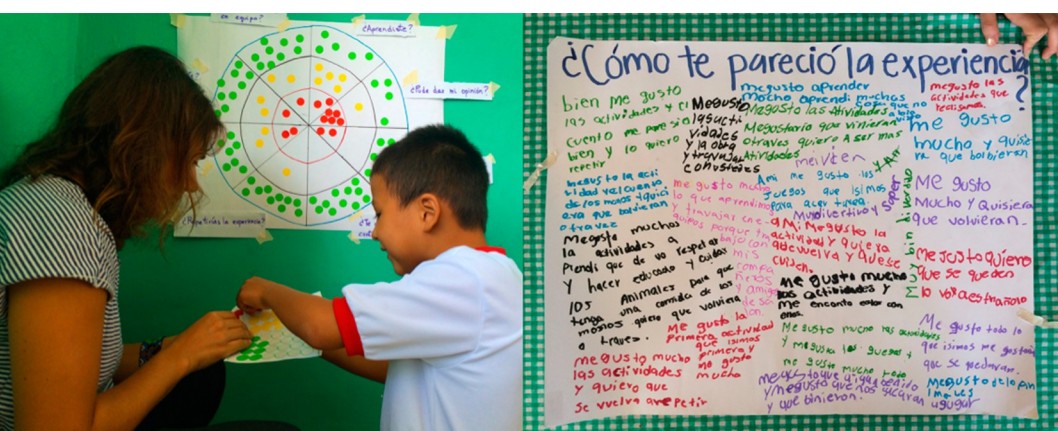

**Fig 3.** *Left* Children's assessment of our performance with a dartboard, showing what they liked and did not like. *Right* Writing space for kids to provide feedback on the activities.

the control group (Intercept), Urban effect is compared with Rural context. Significant results are highlighted in bold.

### Learning indicators, decrease of misconceptions

a. ***Spondias mombin*: Specific item from the black howler monkey's diet.**
   *Jobo* (*Spondias mombin*) appeared for the first time in the post evaluation; it was present in

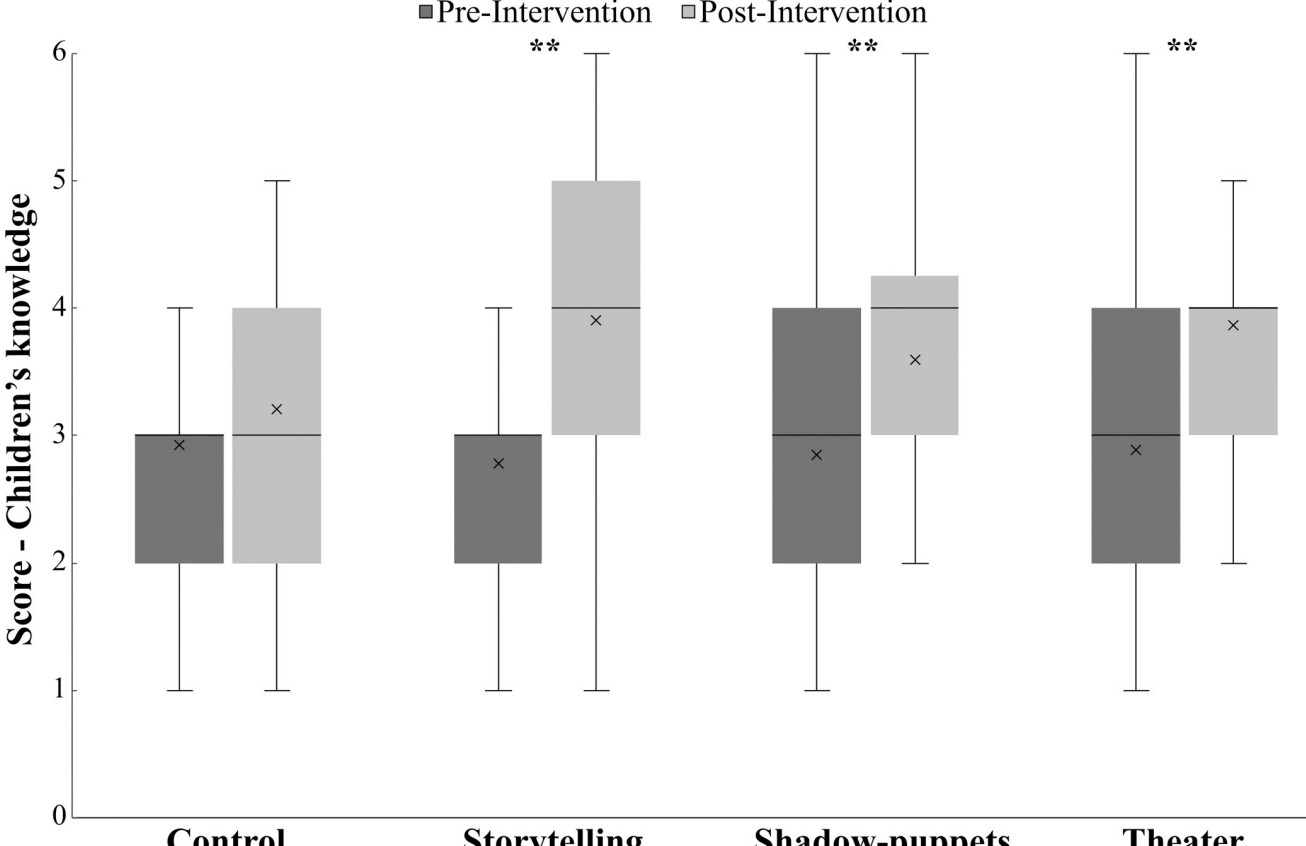

**Fig 4. Changes in children's knowledge between pre-and-post intervention with each arts-based educational techniques and control group.** Lines outside the box represent min and max values. Within the box, the X symbol represents the median and horizontal lines represents the mean.

**Table 2. Results of the GLMM examining the effectiveness of each technique on children's learning about howler monkey behavior, ecology, and conservation at schools from Mexico.**

| Predictor | β | SE | Z | P |
|---|---|---|---|---|
| Intercept | 1.12 | 0.83 | 13.59 | <0.001 |
| Storytelling | 0.23 | 0.10 | 2.33 | **0.002** |
| Shadow-Puppets | 0.13 | 0.10 | 1.25 | 0.210 |
| Theater | 0.18 | 0.10 | 1.75 | 0.080 |
| Urban | 0.14 | 0.07 | 2.01 | **0.045** |

53.7% of the drawings and was influenced by the technique that was used. More children (67.9%) with the theater technique included the *jobo* in their drawings as part of the diet of howler monkeys.

b. **Decrease in the frequency of wrong preconceptions**

Some preconceptions were found in the pre-evaluation [14], but decreased during the post-evaluation, such as bananas as food (from 69.1% to 30.9%) black howlers on the forest floor (from 59.8 to 40.2%), and howlers with brown colored fur (from 57.8 to 42.2%).

## Children's perceptions and feelings

Regarding the children's feedback about the CE program, a majority of positive answers were obtained for all questions. Broadly, > 70% of the children answered that they felt very satisfied in six of the eight questions. Questions with the highest percentage of positive answers were about: whether they learned something new about black howler monkey conservation (72%), the importance of the topic (84%), their enjoyment during the process (89%), and the educational techniques used (87%). The questions that scored negatively included the duration (8% were not satisfied and 25% had intermediate satisfaction) and expression of their opinions (16% were not satisfied and 25% had intermediate satisfaction; Fig 5).

Most of the messages the children wrote on the board (71.3%) were about what they had liked most. In particularly, they often referred to humorous or positive moments during specific activities, including some of the icebreaker games (e.g. "I liked painting, working with you, dancing and singing, I liked everything"; "I liked to perform the theater, I had a lot of fun and I learnt a lot"; "I liked when we saw the show of the monkey and the boy" or "I liked the shadow puppet"). Most of the messages (65.2%) were related to positive feelings and emotions, and some messages were requests to continue our work with them ("I liked doing activities with you, I hope you come back"). Some sentences (40.6%) were linked to the black howler monkeys and what the children learned about them ("I liked to perform the theater, to learn about monkeys, and to draw them").

## Discussion

### Quality of education and conservation programs: The need for innovation in methodologies

Our study examined the effectiveness of different arts-based educational techniques in transmitting the message of the importance to conserve primates and their natural habitat. We found that the educational techniques used improved children's knowledge, with storytelling being the most effective technique for children's learning. Additionally, our aim was to determine the best ways to have a positive educational experience with participatory and inclusive methods, as well as paying special attention to the connection that children have with nature,

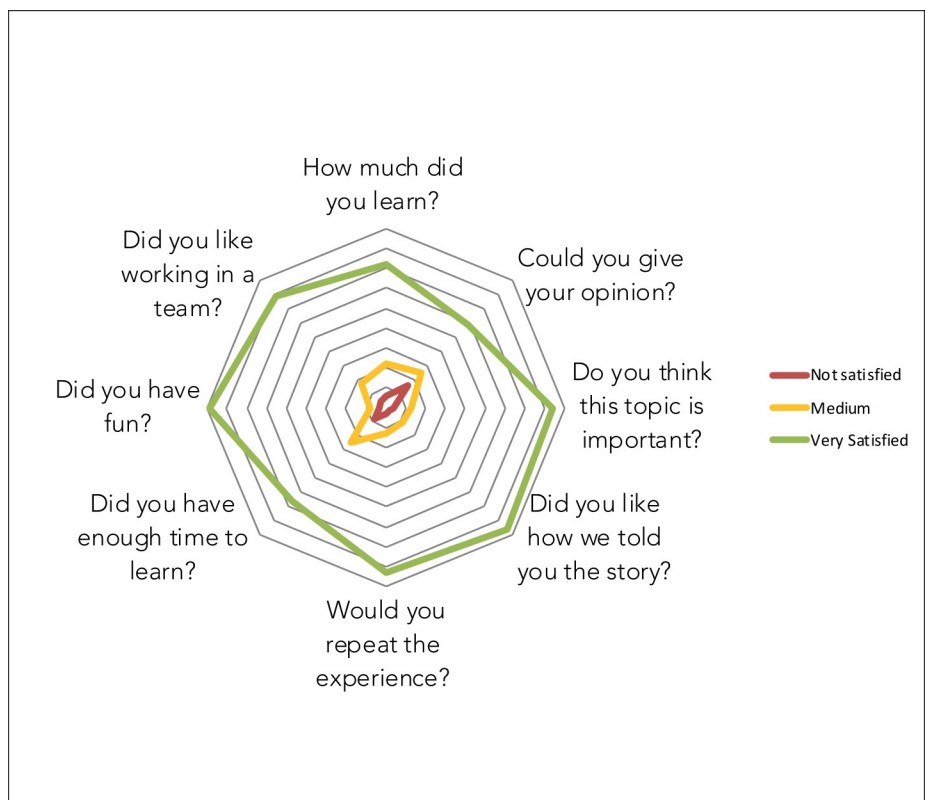

**Fig 5. Children's feedback through answering eight questions about our performance, n = 224.**

or in this case, the black howler monkeys. In Mexico, Education for Sustainability Development (ESD) is still relegated to a minor role in school curricula [40], our work therefore focuses on highlighting an alternative approach for implementing ESD projects in formal schooling. Most Mexican schools teach about ecological aspects (e.g. recycling and water care) in a way that is not always directly linked to local needs, biological conservation, and/or land use [27]. Holistic approaches are needed, where environmental content is linked to individual, collective and global trends and lifestyles. Hence, from our experience, we identified a gap in the teaching field of ESD in Mexico: the species conservation domain. Additionally, it seems that children are primarily following rules rather than being involved in meaningful projects [33]. With the arts-based education approach we were able to encourage critical thinking and potentially promote responsible actions based on autonomous decisions for the conservation of Mexican primates and their habitats.

Nevertheless, it is difficult to determine if the increase in knowledge, decrease of misconceptions, and more positive attitudes towards species conservation can lead to behavioral changes congruent with primate conservation. Considering the time frame of this study, we were unable to measure the impact of this program on potential behavioral changes. However, after the study period, an NGO named *Miku Conservación* was created with the aim to continue the educational and conservational work. As a result of the follow-up visits (three per school during the next two years), some schools started to take autonomous decisions and actions with respect to the conservation of primates and sustainable practices. According to Chawla and Flanders Cushing [41], most studies that show an improvement in young people's responsible environmental behaviors also include an action component, such as starting

community projects and seek strategies to address local socioenvironmental problems. For example, schools were involved in the creation of as school orchards, community plant nurseries, reducing the burning of waste, dedicating some weeks to certain species awareness (Franquesa-Soler, M. unpublished data, December). Long-term programs are more likely to lead to an actual behavioral change [42,43]. The project is currently being extended through the jointly actions of *Miku Conservación*, local school teachers, stakeholders and students.

All these processes demand time, not only for extending the program, but personal time is also needed to realize that certain habits need to change, recognize the roots of the problems and finally decide to change those habits. Most of these habits are related to societal expectations and culture [44]. This is the reason why evaluation is crucial for truly improving subsequent educational and conservational programs. Hence, in the current projects we are focusing on co-researching with the school and the rest of the community by including action components on local issues related to primate conservation and natural resources [45].

In Mexico, it is common for educators–often with limited resources and disposal–to simply hand down the knowledge, and students do not play an active role in the learning process [40,46]. Working on primate conservation allowed for the establishment of real connections between the issues and threats facing primate populations and issues important to the local community issues in ways that are not easy to teach using traditional methods.

## Context and technique predicted children's knowledge on the black howler monkeys

In line with our predictions, we found that children's knowledge about black howler monkeys was affected by the technique used (storytelling, shadow puppet, and theater), being storytelling the most effective one. In contrast to other studies, children's knowledge score was not predicted by gender [47, 48, 49]. We had similar results in the pre-evaluation, in which some categories boys had more knowledge about black howler monkeys than girls [14], but with the implementation of the program these differences disappeared. These results support the importance of designing projects that consider the inclusivity of gender by providing spaces where girls can express themselves and learn the same way that boys do. This design feature is important in Mexico, where gender inequalities still exist, especially in rural communities [50]. It is also particularly significant for Conservation Education (CE) because a positive correlation between gender equality and sustainable development has been observed in countries such as Nepal and India [51].

Knowledge about biodiversity depends on the place of residence [52], and urban residents showed greater concern for the environment [53, 54]. Other recent studies, conversely, indicate that differences among rural and urban citizens may be weakening due to (1) an increase in environmental services in rural areas [55], (2) the migration of urban citizens with positive environmental values to rural communities [56], and (3) the decrease of economic dependence on natural resources by rural communities. In a related study to the present one, we found differences in the pre-evaluation [14]: rural children had greater knowledge about black howler monkeys because they had more opportunities to see them. However, in the post-evaluation of this study we found a reverse pattern: children from urban context scored higher in the evaluation. This can be read as a positive effect, by equating knowledge between both contexts and even benefiting the learning process of students from urban schools.

## Technique used

Except for the control group, all educational techniques (storytelling, shadow puppets and theater), increased children's knowledge score about the black howler monkeys. These findings

confirm the general success of the arts-based educational techniques. It is important to assume that CE programs do not work everywhere in the same way. Accordingly, it is important to test different educational techniques to include cultural factors, gender diversity, and different learning styles. For instance, applying educational techniques (including evaluations) that work for an audience from a high-income country might not necessarily lead to the expected change in a low-income country [4]. In Mexico, ESD is poorly implemented in the school curricula, and arts education is rarely taught [25]. The arts allow children to learn by using all the senses and promote the use of multiple types of intelligences [18]. Additionally, southern Mexico is home to at least 16 indigenous languages [57], and while the majority of schools are not indigenous, reading and writing difficulties in the Spanish language persist. The underlying problem is that the curriculum is the same for the whole country and the assessment instruments are in Spanish [58]. Also, teaching is typically rote-learning (memorization of information based on repetition) instead of promoting educator-learner interactions using creative educational techniques [59,60]. Hence, before applying and evaluating an education program it is crucial to adapt the program's contents and the type of evaluation questions to the local context, as well as considering a variety of different learners [49].

Although all educational techniques successfully increased children's knowledge, storytelling stands out as being the technique that produced the highest score. This matches with the results from the pre-program when learning preferences were evaluated in the same group of children [46]. The pre-program study revealed that the auditory learning style was the preferred one for the children (46.4%). This is consistent with Dunn & Dunn's [61] and Barbe & Milone's [62] results, which showed that it is more likely for children in primary school to learn and retain information through the auditory sense. Here designed tools which considered the children's preference, invited them to reflect and participate actively, and used a technique that was important in the Mayan cultural context [63]. Furthermore, current studies show how narrative and storytelling can facilitate science communication to nonexperts [64], expanding the context of "framing" as being an important component of public outreach [65].

Stories form a link between our imagination and our environment [66]. In fact, for many cultures (e.g. Mayan culture) oral traditions and stories convey local knowledge and wisdom about the environment and our relationship with the earth and with others [63]. Narratives allow the audience to get to know the characters, see different perspectives, and experience their emotions and their environments. Furthermore, it is a good method to achieve emotional connections and symbolic thinking, goals which technology sometimes cannot accomplish. For example, replacing free play and storytelling with the audiovisual system undermines the symbolic-metaphorical intelligence of children [67]. Excessive audiovisual activities isolate children from both the world of imagination and the natural environment. However, during free play, theater, storytelling and similar activities, children can develop inner images. This forms the foundation of future symbolic and metaphorical thinking, as well as concrete operational thinking, such as mathematics, science, philosophy and all forms of knowledge considered as higher education [67].

Our study shows that considering multiple intelligences and senses can improve the effectiveness of CE programs. Not only the intervention, but also the evaluation should also be inclusive to all learners and indigenous cultures, since in primate CE assessment is commonly done through questionnaires [4,29,60]. Drawings have been an excellent tool for evaluating the effectiveness of CE programs, as well as an artistic approach that evaluated the children's knowledge on black howler monkeys.

## Learning indicators, decrease of misconceptions

During the pre-evaluation, we detected the more common misconceptions or assumptions that children showed about black howler monkeys [14]. Long-term studies reveal that black howler monkeys have a highly varied diet, largely dependent on the availability of preferred foods [5]. Therefore, the *jobo* fruit was selected for being representative of both the black howler monkey diet and the southern Mexico region. During the post-evaluation, *jobo* appeared in more than half of the drawings (during the pre-evaluation none), indicating that the educational techniques were successful, the story was understood and the children learned about one of the elements of the actual diet of black howler monkeys.

When analyzing the educational techniques used, theater was the most effective technique for children that incorporated *jobos* into their drawings and, at the same time, reduced the misconception of bananas being a black howler monkey food source. It is important to not only show that some elements are not part of their diet, but also to show which elements are correct. The same principle applies to other pro-environmental behaviors, where it does not help to say what should and should not be done, but instead to show in a simple way what can be contributed [68]. During the theater, children played with props, sometimes simulating the *jobo*. They integrated it into the theatre play, they ate the *jobo* and defecated it to represent the role of black howler monkeys as seed dispersers. The action of touching and using it in a specific situation might serve to materialize and internalize it as a *concrete manipulative* [69]. According to Carbonneau et al. [69], concrete manipulatives make easy learning by (a) promoting abstract reasoning [70], (b) enhancing thought-provoking learners' real-world knowledge [71], (c) allowing the learner to apply the concept for improved encoding [72], and (d) offering chances for learner-driven exploration while discovering new concepts [73].

Assumptions or misconceptions decreased in the post-evaluation analysis, indicating that our educational techniques not only help increasing learning, but also to clarify some children's sources of confusion. The origin of the myth that monkeys naturally eat bananas remains unidentified, but its persistence is a misconception that primate conservation education programs should address as it tends to anthropomorphize monkeys. The banana concept was tackled through the help of "dislike" sounds and negative expressions during the educational techniques. The brown color was explained in the stories with the different formats, but it was also reinforced by showing a real picture of the animal at the end of the educational technique. The forest floor misconception was highlighted in the story by relaying that howler monkeys arrived to an extremely deforested area and needed to cross the highway on the ground to access to other sources of fruit, but that some died while trying. Other threats and diseases that these animals face on the forest floor were also shown.

## Satisfaction survey

The use of evaluation tools is really important for CE programs [60]. To understand which conservation programs are effective, it is essential to assess the children's perceptions and knowledge about certain topics. It is also important to understand the children's feelings and opinions about the conservation program [4, 29]. We sought to assess the latter through a satisfaction survey. Because children from Mexican communities are usually taught within the traditional teaching system, it is not common for them to be consulted about the educational programs that they receive or to be asked to evaluate their teachers' performance. For this reason, it is important to take into account that their answers may be biased, resulting in overestimating our performance. However, the satisfaction survey is still a useful tool to explore the children's reactions to our activities and to enhance critical thinking, which is necessary for the decision-making processes regarding sustainability.

The general results of the satisfaction survey show a positive response to this conservation education program. Children stated that they had learned about black howler monkey conservation and considered it an important issue. We attribute these reactions to their enjoyment of the learning process and the arts-based educational techniques used during the program. Including an emotional component in the design of education programs can act as a motivator by facilitating students' engagement and making the learning process more stimulating [22]. Some children thought that the time invested in the implementation of the program should have been longer. This is in agreement with some primate education projects that have found that longer programs are associated with a greater increase in participants' knowledge [60,28] and that long-term projects can be more effective [42]. That being said, other studies found that the length of participant involvement did not affect knowledge retention [12,74]. All the message posts left by the children on the board contained positive content. Most of these messages allude to their enjoyment with some of the artistic activities or icebreaker games we did during the process. On one hand, the use of games is a powerful teaching strategy because it makes the learning process more interesting and fun [75]. On the other hand, researchers have found that arts offer a way for people to connect emotionally to the conservation topic of interest and are therefore proving successful. A good learning process includes feelings, which is vital to achieving long-term changes in perceptions and behaviors [22]. Moreover, children showed in their messages that they really appreciated our activities and the time they spent with us, and that they would love to repeat the experience.

## Final remarks

We contend that the program was effective not only because of the educational techniques used, but because it was a holistic program. [76]. Since we started asking for permits from the *Secretaría de Educación Pública*—Secretariat of Public Education in the Government office of each State, we considered presentation days, ice breaker games, establishing rapport or trust with the school community (and the general community). We performed evaluations with inclusive, qualitative and comfortable evaluating tools (drawings). We worked with small groups, naming those groups with the corresponding fruits and animals which appeared in the story. We conducted activities beforehand in order to prepare the artistic language for the different educational techniques. We based the program on an artistic approach, creating spaces for dialogue, and other drawing activities to reinforce learning. We also conducted the satisfaction survey to give children a voice, as well as games to invite reflection and the sharing of experiences, emotions and cultural exchanges. We spent 2–3 weeks at each place, living there, sharing food with the local people, getting to know the parents, grandparents and places in which they loved to play or explore after school. We also participated in some activities important for the local community (e.g. town or school festivals, sport events). To sum up, it was a holistic experience. The reinforcement of this work and relation continues through the NGO, where some schools and communities from the initial program have already started to have autonomous decisions and actions seeking species conservation and sustainable development. Holistic experiences involve conveying a complete idea or story within the educational context. They thus carry high potential to provide a coherent picture of the relevance of the educational technique and a clear take-home point for students to reflect upon or pursue following the experience [76].

## Supporting information

**S1 Table. Data set underlying the study of evidence-based conservation education in Mexican communities: Connecting arts and science.** The dependent variables are the different

factors considered during the implementation of the educational intervention (column 2–7) the independent variable is the Score (column 8).
(DOCX)

## Acknowledgments

The authors are grateful to the participating schools and the rest of the communities. We would like to express our deepest gratitude to Erin P. Riley and Maria Lay for their valuable suggestions to the structure of the manuscript, language editing, and proofreading. Thank you to Ellen Andresen, Sergi López-Torres and Denise Spaan for English language editing and valuable comments. Thanks to Esther Castro, Laura Jayme for their support during fieldwork and artistic activities, and Adriana Sandoval-Comte who helped with mapping. We also thank Jeffrey Sayer and another anonymous reviewer for their contributions to improve this manuscript.

## Author Contributions

**Conceptualization:** Montserrat Franquesa-Soler, Lucía Jorge-Sales, Patricia Moreno-Casasola, Juan Carlos Serio-Silva.

**Data curation:** Montserrat Franquesa-Soler, Lucía Jorge-Sales, John F. Aristizabal.

**Formal analysis:** Montserrat Franquesa-Soler, Lucía Jorge-Sales, John F. Aristizabal.

**Funding acquisition:** Juan Carlos Serio-Silva.

**Investigation:** Montserrat Franquesa-Soler.

**Methodology:** Montserrat Franquesa-Soler, John F. Aristizabal.

**Project administration:** Juan Carlos Serio-Silva.

**Software:** John F. Aristizabal.

**Supervision:** Patricia Moreno-Casasola, Juan Carlos Serio-Silva.

**Visualization:** Montserrat Franquesa-Soler.

**Writing – original draft:** Montserrat Franquesa-Soler, John F. Aristizabal.

**Writing – review & editing:** Montserrat Franquesa-Soler, Lucía Jorge-Sales, John F. Aristizabal, Patricia Moreno-Casasola, Juan Carlos Serio-Silva.

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
