## [Decision Letter · Decision Letter 0]

12 Nov 2019

PONE-D-19-15156

Evidence-based conservation education in Mexican communities: connecting arts and science

PLOS ONE

Dear Dr. Franquesa-Soler,

Thank you for submitting your manuscript to PLOS ONE. After careful consideration, we feel that it has merit but does not fully meet PLOS ONE’s publication criteria as it currently stands. Therefore, we invite you to submit a revised version of the manuscript that addresses the points raised during the review process.

Both reviewers and this editor share the same understanding:

The main required changes are related to the connection between the educational actions and behavioral changes in the students. In this regard, please, make clear this issue in the Discussion and carefully consider all the remarks made by reviewer 2.

Likewise, it is necessary to bring more evidence that the behavioral changes were actually due to the performed activities and not due to other causes.

We would appreciate receiving your revised manuscript by Dec 27 2019 11:59PM. To enhance the reproducibility of your results, we recommend that if applicable you deposit your laboratory protocols in protocols.io, where a protocol can be assigned its own identifier (DOI) such that it can be cited independently in the future. For instructions see: http://journals.plos.org/plosone/s/submission-guidelines#loc-laboratory-protocols

We look forward to receiving your revised manuscript.

Kind regards,

Paulo Takeo Sano, Ph.D.

Academic Editor

PLOS ONE

Additional Editor Comments (if provided):

The study obtained the official permission to research in schools in Mexico; however, the manuscript does not inform all other necessary ethical procedures.

PlosOne Ethical Statement requires an approval number and/or statement indicating approval to the research (not only the official, but the Ethical Committee approval, as it is a customary practice in such surveys). It is also necessary to indicate the form of consent obtained (written/oral).

There is no mention to an Informed Consent Form (ICF) and an Ethical Committee aproval. An ICF has to be aproved by the Ethical Committee and subsequently signed by each of the trial participant or at least one legally authorised representative, since they are children.

2) You indicated that you had ethical approval for your study. In your Methods section, please ensure you have also stated whether you obtained consent from parents or guardians of the minors included in the study or whether the research ethics committee or IRB specifically waived the need for their consent.

3) Thank you for including your ethics statement : "All primary schools agreed to join our educational intervention and official permissions were obtained by Institutional review boards of School Council and Secretaria de Educación Pública - Secretariat of Public Education- in the Governement office of each state (Documents attached as supplementary material)".   

a.Please amend your current ethics statement to confirm that your named institutional review board or ethics committee specifically approved this study.

b.Once you have amended this/these statement(s) in the Methods section of the manuscript, please add the same text to the “Ethics Statement” field of the submission form (via “Edit Submission”).

4)  Please expand the acronym JCSS (as indicated in your financial disclosure) so that it states the name of your funders in full.

5) Please remove your figures from within your manuscript file, leaving only the individual TIFF/EPS image files, uploaded separately.  These will be automatically included in the reviewers’ PDF.

6) We note that Figure 2 includes an image of a participant in the study.

7) In your Data Availability statement, you have not specified where the minimal data set underlying the results described in your manuscript can be found. PLOS defines a study's minimal data set as the underlying data used to reach the conclusions drawn in the manuscript and any additional data required to replicate the reported study findings in their entirety. All PLOS journals require that the minimal data set be made fully available. For more information about our data policy, please see http://journals.plos.org/plosone/s/data-availability.

Reviewers' comments:

Reviewer's Responses to Questions

**Comments to the Author**

1. Is the manuscript technically sound, and do the data support the conclusions?

Reviewer #1: Yes

Reviewer #2: Partly

2. Has the statistical analysis been performed appropriately and rigorously? 

Reviewer #1: I Don't Know

Reviewer #2: I Don't Know

3. Have the authors made all data underlying the findings in their manuscript fully available?

Reviewer #1: Yes

Reviewer #2: Yes

4. Is the manuscript presented in an intelligible fashion and written in standard English?

Reviewer #1: Yes

Reviewer #2: No

5. Review Comments to the Author

Reviewer #1: The paper makes a useful contribution to knowledge of the impact of conservation education programs. The data does not provide any information on the changes in behavior of participants as a result of the educational activities and this should be made clear in the conclusions. The English is in general pretty good but there are some words used wrongly and there are one or two problems of word order - some only of these have been corrected in the attached ms. I Recommend that the authors seek help with copy editing.

Reviewer #2: This paper is an attempt to highlight the importance of conservation education for the management of an endangered primate in Mexico. The authors describe, in detail, the various methods and approaches used by the research team to understand and influence? understanding of the conservation value of the black howler monkey. However, it is not clear from the outset what the research itself is trying to achieve. While understanding perceptions of current knowledge of schoolchildren is one way to focus on conservation education, it is unclear how that will relate to actual behavioural change that will lead to better conservation outcomes on the ground. Indeed these two processes are worryingly disconnected as presented in the paper. This reviewer is unsure whether a child understands that the main diet of the black howler monkey are the fruits of Spondias mombin and not bananas will lead to changes in behaviour related to the conservation of the primate itself. There is a very strong focus in the paper on methods, approaches and long discourse on educational techniques (indeed perhaps this could be reduced somewhat) but very little in what the research is hoping to achieve in terms of conservation outcomes. Perhaps this could be made more explicit.

As it reads, the paper could really do with a thorough review by a native English speaker. There are numerous instances of poor and confusing grammar, superfluous information (the first sentence of the abstract is one example), unnecessary use of capitals throughout and the introduction of some clang e.g. "kids". This will need to be amended prior to publication.

In short, this paper could provide some insights into conservation education methods and approaches as long as there is an explicit link to overall conservation outcome, real or perceived. A shorter, concise and focused version of this paper would perhaps work in this regard.

6. PLOS authors have the option to publish the peer review history of their article (what does this mean?). If published, this will include your full peer review and any attached files.

Reviewer #1: Yes: Jeffrey Sayer

Reviewer #2: No

---

## [Author Response · Author response to Decision Letter 0]

27 Dec 2019

R/ Thanks for the opportunity to revise and resubmit our manuscript. We hope our revisions solve all of their main concerns, and that the same reviewers will agree to review the new version of our manuscript. None of the authors is a native-English speaker, hence you may have found some grammar errors in this document or the cover letter which was not proofread by anyone else, but the manuscript was reviewed by two colleagues who are native English speakers. Also, the whole article has been reviewed and improved in Abstract, Introduction, Analysis, Results and Discussion by two colleagues, one of them is now a new co-author for his important contribution on improving the manuscript. Also, all figures were verified through PACE to ensure that our figures meet PLOS one requirements.

The main required changes are related to the connection between the educational actions and behavioral changes in the students. In this regard, please, make clear this issue in the Discussion and carefully consider all the remarks made by reviewer 2. Likewise, it is necessary to bring more evidence that the behavioral changes were actually due to the performed activities and not due to other causes.

R/ This concern have been attended and included in the discussion 344-360.

The study obtained the official permission to research in schools in Mexico; however, the manuscript does not inform all other necessary ethical procedures. PlosOne Ethical Statement requires an approval number and/or statement indicating approval to the research (not only the official, but the Ethical Committee approval, as it is a customary practice in such surveys). 

 It is also necessary to indicate the form of consent obtained (written/oral). There is no mention to an Informed Consent Form (ICF) and an Ethical Committee aproval. An ICF has to be aproved by the Ethical Committee and subsequently signed by each of the trial participant or at least one legally authorised representative, since they are children.

R/ We indicated in Methods section that we obtained Informed oral Consent and the Ethical Committee approval, aside from the official permission to research in public schools in Mexico.

2) You indicated that you had ethical approval for your study. In your Methods section, please ensure you have also stated whether you obtained consent from parents or guardians of the minors included in the study or whether the research ethics committee or IRB specifically waived the need for their consent.

R/ We indicated in Methods section that we obtained Informed oral Consent and the Ethical Committee approval, aside from the official permission to research in public schools in Mexico.

3) Thank you for including your ethics statement : "All primary schools agreed to join our educational intervention and official permissions were obtained by Institutional review boards of School Council and Secretaria de Educación Pública - Secretariat of Public Education- in the Governement office of each state (Documents attached as supplementary material)". 

a. Please amend your current ethics statement to confirm that your named institutional review board or ethics committee specifically approved this study.

R/ The Ethical Committee of our Institution approved the present study, we stated this in the Methods section and we attached the official letter signed for this revised version.

R/ We also added the same text to the “Ethics Statement” field of the submission form via “Edit Submission”

4) Please expand the acronym JCSS (as indicated in your financial disclosure) so that it states the name of your funders in full.

R/ Done

R/ Done

5) Please remove your figures from within your manuscript file, leaving only the individual TIFF/EPS image files, uploaded separately. These will be automatically included in the reviewers’ PDF.

R/ Done

6) We note that Figure 2 includes an image of a participant in the study.

R/ We obtained the parents’ permission to disseminate the pictures of their sons and daughters through oral and written consent, we attached the format that we used during the project and each tutor signed. This picture is quite representative of how the process was instead of showing only the dart board itself. Besides, we did not reveal any personal information of the participant and the capture is from behind, it is barely impossible to know the identity of the participant. But we agree to change the Figure 3 if the reasons explained above (and the formats we used) are not enough to meet the criteria of Plos One publication.

7) In your Data Availability statement, you have not specified where the minimal data set underlying the results described in your manuscript can be found. PLOS defines a study's minimal data set as the underlying data used to reach the conclusions drawn in the manuscript and any additional data required to replicate the reported study findings in their entirety. All PLOS journals require that the minimal data set be made fully available. For more information about our data policy, please see http://journals.plos.org/plosone/s/data-availability.

Comments to the Author

1. Is the manuscript technically sound, and do the data support the conclusions?

Reviewer #1: Yes

Reviewer #2: Partly

R/ We aim that with the modifications to statistics, analysis, descriptions in methods and providing new information regarding concerns that reviewers showed, the second reviewer could be more satisfied with the data supporting conclusions in this new version. 

Additionally, the program was designed with a multidisciplinary approach, approved by biologists, theater artists, teachers and psychologists. Also, the pilot test was conducted in other schools to test and fine-tune the instruments. Everything was repeated respecting scientific rigor, using the same educational facilitators, using the same premises, questions, activities, etc. Being a socio-environmental research with children it is difficult to maintain the same experimental conditions as a laboratory work, but this study followed a method as solid as possible.

2. Has the statistical analysis been performed appropriately and rigorously?

Reviewer #1: I Don't Know

Reviewer #2: I Don't Know

R / Given the response of the reviewers, we decided to rethink and substantially improve the statistical analyzes (now GLM; see Data analysis section) with the collaboration of an expert in the treatment of the data. Additionally, we present a new box-figure (fig. 4) that show medians and variance measures (min and max values). We also improved the methods section with a more detailed and clearer description of analysis performed. For this reason, in the move one of the researchers from Acknowledgments to a new Co-author for his important contribution, that generally improve the version of the article and meet the suggestions of the reviewers.

3. Have the authors made all data underlying the findings in their manuscript fully available?

Reviewer #1: Yes

Reviewer #2: Yes

4. Is the manuscript presented in an intelligible fashion and written in standard English?

Reviewer #1: Yes

Reviewer #2: No

R/ Following both Reviewers’ suggestions, we asked for help with copy editing. Then, the new version of the article was carefully reviewed by a native English socio-environmental researcher to improve the redaction of the manuscript. Figure 5 also have been edited and therefore we attached another version in the resubmission.

5. Review Comments to the Author

Reviewer #1: The paper makes a useful contribution to knowledge of the impact of conservation education programs. The data does not provide any information on the changes in behavior of participants as a result of the educational activities and this should be made clear in the conclusions. The English is in general pretty good but there are some words used wrongly and there are one or two problems of word order - some only of these have been corrected in the attached ms. I Recommend that the authors seek help with copy editing.

R/ We appreciate you find the article interesting and suitable for Plos One public. We also appreciate and agreed with your comments and we discussed about this connection between the educational programs and the chances in behavior. Also, a native speaker researcher helped with copy editing, so hopefully this new version meets the journal’s standards.

Reviewer #2: This paper is an attempt to highlight the importance of conservation education for the management of an endangered primate in Mexico. The authors describe, in detail, the various methods and approaches used by the research team to understand and influence? understanding of the conservation value of the black howler monkey. However, it is not clear from the outset what the research itself is trying to achieve. While understanding perceptions of current knowledge of schoolchildren is one way to focus on conservation education, it is unclear how that will relate to actual behavioural change that will lead to better conservation outcomes on the ground. Indeed these two processes are worryingly disconnected as presented in the paper. This reviewer is unsure whether a child understands that the main diet of the black howler monkey are the fruits of Spondias mombin and not bananas will lead to changes in behaviour related to the conservation of the primate itself. There is a very strong focus in the paper on methods, approaches and long discourse on educational techniques (indeed perhaps this could be reduced somewhat) but very little in what the research is hoping to achieve in terms of conservation outcomes. Perhaps this could be made more explicit. As it reads, the paper could really do with a thorough review by a native English speaker. There are numerous instances of poor and confusing grammar, superfluous information (the first sentence of the abstract is one example), unnecessary use of capitals throughout and the introduction of some clang e.g. "kids". This will need to be amended prior to publication.

In short, this paper could provide some insights into conservation education methods and approaches as long as there is an explicit link to overall conservation outcome, real or perceived. A shorter, concise and focused version of this paper would perhaps work in this regard.

6. PLOS authors have the option to publish the peer review history of their article (what does this mean?). If published, this will include your full peer review and any attached files.

Do you want your identity to be public for this peer review? For information about this choice, including consent withdrawal, please see our Privacy Policy.

Reviewer #1: Yes: Jeffrey Sayer

Reviewer #2: No

---

## [Editor Report · Decision Letter 1]

15 Jan 2020

Evidence-based conservation education in Mexican communities: connecting arts and science

PONE-D-19-15156R1

Dear Dr. Franquesa-Soler,

We are pleased to inform you that your manuscript has been judged scientifically suitable for publication and will be formally accepted for publication once it complies with all outstanding technical requirements.

With kind regards,

Andrew R. Dalby, PhD

Academic Editor

PLOS ONE
---

## [Editor Report · Acceptance letter]

23 Jan 2020

PONE-D-19-15156R1 

Evidence-based conservation education in Mexican communities: connecting arts and science 

Dear Dr. Franquesa-Soler:

I am pleased to inform you that your manuscript has been deemed suitable for publication in PLOS ONE. Congratulations! Your manuscript is now with our production department. 

With kind regards,

on behalf of

Dr. Andrew R. Dalby 

Academic Editor

PLOS ONE